# The Polyvalent Role of CD30 for Cancer Diagnosis and Treatment

**DOI:** 10.3390/cells12131783

**Published:** 2023-07-05

**Authors:** Adrian Vasile Dumitru, Dana Antonia Țăpoi, Georgian Halcu, Octavian Munteanu, David-Ioan Dumitrascu, Mihail Constantin Ceaușu, Ancuța-Augustina Gheorghișan-Gălățeanu

**Affiliations:** 1Department of Pathology, Carol Davila University of Medicine and Pharmacy, 020021 Bucharest, Romania; vasile.dumitru@umfcd.ro (A.V.D.); ceausu_mihai@yahoo.com (M.C.C.); 2Department of Pathology, University Emergency Hospital, 050098 Bucharest, Romania; 3Department of Pathology, Colțea Clinical Hospital, 030171 Bucharest, Romania; 4Department of Anatomy, Carol Davila University of Medicine and Pharmacy, 020021 Bucharest, Romania; octav_munteanu@yahoo.com; 5Department of Obstetrics and Gynecology, University Emergency Hospital, 050098 Bucharest, Romania; 6Faculty of Medicine, Carol Davila University of Medicine and Pharmacy, 020021 Bucharest, Romania; david-ioan.dumitrascu0720@stud.umfcd.ro; 7Department of Pathology, Alexandru Trestioreanu Institute of Oncology, 022328 Bucharest, Romania; 8Department of Cellular and Molecular Biology and Histology, Carol Davila University of Medicine and Pharmacy, 020021 Bucharest, Romania; ancuta.gheorghisan@umfcd.ro; 9C.I. Parhon National Institute of Endocrinology, 011863 Bucharest, Romania

**Keywords:** CD30, anaplastic large cell lymphoma, embryonal carcinoma, Hodgkin lymphoma, cancer treatment, tumor marker

## Abstract

CD30, also known as TNFRSF8 (tumor necrosis factor receptor superfamily member 8), is a protein receptor that is heavily glycosylated inside the Golgi apparatus, as well as a tumor marker that is found on the surface of specific cells in the body, including certain immune cells and cancer ones. This review aims to shed light on the critical importance of CD30, from its emergence in the cell to its position in diagnosing various diseases, including Hodgkin lymphoma, where it is expressed on Hodgkin and Reed–Sternberg cells, as well as embryonal carcinoma, anaplastic large cell lymphoma (ALCL), and cutaneous T-cell lymphoma (CTCL). In addition to its role in positive diagnosis, targeting CD30 has been a promising approach treating CD30-positive lymphomas, and there is ongoing research into the potential use of CD30-targeted therapies for autoimmune disorders. We aim to elaborate on CD30’s roles as a tumor marker, supporting thus the hypothesis that this receptor might be the aim of cytostatic treatment.

## 1. Introduction

CD30 (TNFRSF8) is a protein receptor located on the surface of cells and belongs to the tumor necrosis factor receptor superfamily. It can be found on in multiple cell types, including activated T cells, B cells, natural killer (NK) cells, and lymphoid cells. CD30 engages in various biological processes by interacting with its ligand CD30L (also known as CD153). This interaction gives rise to numerous functions, as described below.

CD30 is a transmembrane protein with an approximate molecular weight of 120 kDa and is classified as a type I protein. Its extracellular domain features six cysteine-rich pseudo repeat motifs; within its cytoplasmic tail, multiple sequences exist that are capable of binding to TNF receptor-associated factors (TRAFs). These TRAF-binding sequences facilitate the activation of various signaling pathways, including nuclear factor kappa B (NF-kB) and extracellular signal-regulated kinase (ERK) pathways. Notably, overexpression of CD30 has been found to cause self-aggregation, which recruits TRAFs for recruitment and activation of NF-kB regardless of CD30L expression levels. When present naturally in individuals without pathologic conditions, CD30 expression is typically limited to activated B and T lymphocytes [1]. CD30L, commonly known as CD153, is a type II transmembrane protein belonging to the TNF superfamily 8 (TNFSF8). The human *CD30L* gene can be found on chromosome 9q33, and CD30L is expressed in both resting and activated B cells, activated T cells, natural killer (NK) cells, eosinophils granulocytes, monocytes, and mast cells. In addition, CD30L can be found in epithelial cells and Hassall’s corpuscles within the thymus medulla, as well as both protein and mRNA forms of CD30L that are expressed in malignant hematopoietic cells, such as chronic lymphocytic leukemia, follicular B-cell lymphoma, hairy cell leukemia, T-cell lymphoblastic lymphoma, and adult T-cell leukemia lymphoma. However, its exact role within healthy individuals remains ill-understood since no human diseases have been associated with defects in either gene [2,3].

Initial identification of CD30 as an antigen found only on Reed–Sternberg cells associated with Hodgkin’s disease led to further investigations, showing that it is also present on activated lymphoid cells and anaplastic large cell lymphomas. This antigen can now be found in tissues such as mesothelium, soft tissue tumors, decidua, and activated macrophages. Studies conducted in vitro indicate that CD30 may play a key role in lymphoid cell signaling pathways related to cell proliferation, apoptosis, and cytotoxicity. Its most prominent non-lymphoid expression can be seen in embryonal carcinoma. Studies have consistently demonstrated the strong expression of CD30 at both protein and mRNA levels in embryonal carcinoma tumors. Lower expression was also detected in seminomas and yolk sac tumors. However, a more recent investigation contradicted these results by finding no CD30 positivity among 27 mixed tumor cases that did not contain embryonal carcinoma elements. Due to an increasing rate of testicular tumors with embryonal elements, such as embryonal carcinomas, it has become more important than ever before to detect embryonal antibodies and confirm their presence within nonseminomatous germ cell tumors. Antibodies, such as anti-phospholipid, are also extremely helpful when trying to determine which tumor cells contain embryonal elements and distinguishing embryonal carcinoma from other elements within nonseminomatous germ cell tumors [4,5,6].

Even though CD30 has long been recognized as an essential marker in various lymphomas and activation molecule in B and T cells, its biological role remains unexplored. Researchers wanted to explore the effects of CD30 signaling, so they employed the C10 antibody, an agonist of human CD30, to stimulate CD30 on YT (yolk tumor) cells. Subsequent gene array analysis demonstrated the induction and suppression of approximately 750 gene products and 90 gene products by CD30 signals, showing more than two-fold changes. CD30 signaling involves both TRAF2-dependent and independent pathways. These signals have the ability to reduce the activity of effector cells by modulating gene expression associated with their cytotoxic functions in natural killer (NK) and T cells. At YT, a large granular lymphoma cell line, CD30 signals interfere with expression of mRNA-encoding cytotoxic effector molecules, such as Fas ligand, Perforin, and Granzyme B, which leads to their loss and, consequently, reduces cytotoxicity. CD30 completely inhibits Cellula myc (cmyc), a regulator of proliferation and an upstream regulator of Fas ligand. Furthermore, CD30 induces and upregulates CCR7 expression, suggesting its involvement in lymphocyte trafficking to lymph nodes. CD30 upregulates Fas (TNFRSF6), death receptor 3 (TNFRSF25), and TNF-related apoptosis-inducing ligand (TNFSF10), which indicates increased susceptibility to apoptotic signals. Conversely, upregulation of TNFR-associated factor 1 and cellular inhibitor of apoptosis 2 protects cells against specific forms of apoptosis. These studies demonstrate how CD30 signaling may hinder lymphocyte effector function and proliferation while simultaneously driving them towards lymph nodes and increasing susceptibility to specific apoptotic signals. They offer one potential mechanism behind CD30’s observed suppression of CD8 cytotoxic T lymphocyte (CTL) activity in diabetes models in vivo [5,7,8,9]. The effects of CD30 in oncogenesis are presented in Figure 1.

Understanding CD30 and its functions within various biological processes continues to advance, with research providing new insights and potential therapeutic implications. CD30 has been identified as playing an essential role in the activation and survival of lymphocytes and in modulating immune responses. Researchers are exploring CD30’s role in organizing and maintaining lymphoid tissues, as well as uncovering its significance for life-threatening conditions, such as Hodgkin lymphoma, embryonal carcinoma, anaplastic large cell lymphoma, and cutaneous T-cell lymphoma. Their investigations aim to gain more insight into its mechanisms as well as potential implications for disease management.

## 2. CD30 in Disease

### 2.1. Hodgkin Lymphoma and CD30

Hodgkin lymphoma, characterized by Reed–Sternberg cells, has long been linked with CD30 expression on their surface cells. CD30 plays an integral part in the Hodgkin lymphoma pathology by supporting Reed–Sternberg survival and proliferation; activation via signaling pathway by CD30 triggers the activation of nuclear factor-kappa B (NF-kB), an activator transcription factor responsible for gene expression which promotes survival resistance to apoptosis and ultimately contributes to sustained growth and survival of Reed–Sternberg cells over time.

Lymphomas that affect the thymus can be divided into T-cell and B-cell lineages and Hodgkin lymphomas. Lymphomas involving this organ account for approximately 25% of mediastinal tumors; about 13% are Hodgkin lymphomas (HL), while 13% are non-Hodgkin lymphomas (NHL). Only around 3% and 6%, respectively, arise as primary mediastinal malignancies. Approximately 50% or 20%, respectively, have systemic involvement; mediastinal and hilar lymph nodes are affected in cases with generalized involvement. Currently, two histological subtypes that most frequently affect localized mediastinal involvement are PMBL (primary mediastinal B-cell lymphoma) l and T-lymphoblastic lymphoma (T-Lb). Both may arise from thymic tissue. In addition, age may influence which subtype is diagnosed. Subtype incidence has been discussed extensively [10].

Below, we will examine some of the more prevalent mediastinal/primary thymic lymphomas and their key immunohistochemical markers, notable morphological characteristics, and diagnostic challenges. Furthermore, we will describe rare, tumor-like lesions of particular interest due to their complex pathogenesis, clinical features, and status as specific or uncommon biological entities, such as Castleman disease, in both mediastinum as well as thymus and IgG4-related diseases [11].

Lymphomas of the thymus may encompass numerous subtypes, from immature precursor cells (both T and B cells) to mature peripheral T and B cells. Hodgkin lymphoma of the classical type (cHL), most frequently observed in both pediatric patients and adults, is the most prevalent lymphoma. The nodular sclerosis subtype of Hodgkin lymphoma is especially prevalent here; T-cell precursor type lymphomas tend to be more predominant among young patients compared to lymphomas originating in adults; primary cHL cases that specifically target the thymus are relatively rare in both age groups [12,13].

Thymic lymphomas typically arise from B cells. Thymic B cells tend to reside mainly within the medulla or perivascular spaces and exhibit distinct phenotypic characteristics compared to other subsets of B cells, although their exact relationship remains poorly understood. Thymic B cells may contribute to B-cell thymic lymphoma or classical Hodgkin lymphoma (cHL) of the thymus, among other cancers. Due to neoplastic growth’s destructive nature, biopsies or surgical specimens of lymphoid masses may make it challenging to identify remnants and features indicative of their thymic origin. If these remnants are identified, however, they should be accurately acknowledged as such [14].

The nodular sclerosis (NS) variant of classical Hodgkin lymphomas (cHLs) in the mediastinum are the most prevalent subtype, making up 50–70% of primary mediastinal lymphomas. Thymus-NS lymphoma, in particular, affects young individuals more often than older ones and women more often than men. Under conditions of sclerosis with a polymorphous inflammatory cell population, it is beneficial to identify CD30+ cells, even though they may be sparse in fibrous backgrounds that will aid diagnosis. Reed–Sternberg cells (RS) can be identified by their large size, abundant eosinophilic cytoplasm, two or more nuclei, and an abundance of eosinophilic nucleoli. Lacunar cells (LCs) typically feature small hyperlobulated nuclei with small nucleoli and clear, retractile cytoplasm, and they often represent the NS subtype of chronic herpetic lymphoma (cHL). Furthermore, this subtype can often lead to reactive epithelial cell (EC) proliferation and cystic changes, which can easily resemble a thymoma. An extensive sampling of mediastinal cystic lesions is necessary to identify possible focal sites of chronic lymphocytic leukemia within their walls. Misdiagnosis of chronic lymphocytic leukemia of the thymus as primary mediastinal B-cell lymphoma (PMBL) can be common as both can induce sclerotic reactions and may display RS-like cells that have similar characteristics. Both disorders share B-cell origins and display similar morphologies and clinical presentations [15].

CD30 expression on Reed–Sternberg cells makes it an attractive target for diagnostic and therapeutic uses, including Hodgkin lymphoma treatment. Antibody–drug conjugates specifically targeting CD30, such as brentuximab vedotin, have demonstrated positive outcomes in treatment modalities. These novel therapeutic interventions offer significant promise in managing this disease.

As T-lymphoblastic lymphoma and B1 thymoma share many morphological similarities, it can be difficult to differentiate them using lymphoblasts alone. In addition, both tumors often exhibit frequent mitosis and extensive necrosis, which makes diagnosis even more challenging. Therefore, when dealing with necrotic tumors or small biopsies that cannot provide sufficient tissue samples for diagnosis, relying heavily on specific immunohistochemical markers that target lymphoblastic precursors like LMO2 can often help. Previous research has established that CDK6 staining is limited to T-lymphoblastic lymphoma cells, and subcapsular lymphoblasts present in normal or hyperplastic thymuses. Furthermore, T-lymphoblastic lymphomas tend to exhibit rare remnants and epithelial cell (EC) networks due to the highly destructive nature of tumor growth. In contrast, B1 thymomas may exhibit partial presence of EC networks even within necrotic tumors, which can be detected using appropriate immunohistochemical stains (e.g., cytokeratins). Also, rarely, mature, peripheral T-cell lymphomas have been reported in thymus or mediastinal lymph nodes [16,17]. In Table 1, we compare different aspects of T-lymphoblastic lymphoma and B1 thymoma, highlighting the elements that lead to differential diagnostics.

Almost all the CD30 positive neoplasms are composite tumors in which the CD30 component is at least a minimal part of corresponding tumors. A composite tumor consists of different cell types or components. In the context of CD30-positive neoplasms, this means that the tumors are not solely composed of CD30-positive cells, but also contain other types of cells. For example, in Hodgkin lymphoma, the CD30 component is only a part of the overall tumor, and it may be present in minimal quantities compared to other components of the tumor [18].

Therefore, any diagnostic or predictive speculation based solely on the identification and quantification of CD30-positive cells cannot be reliable. In other words, the presence or number of CD30-positive cells alone cannot provide a definitive diagnosis or predict the behavior of the tumor. Additional morphological (structural) features and further investigations are necessary for accurate diagnosis and prediction, and some of them are highlighted in Table 1.

In addition to conventional histological sections, CD30 positive cells can be identified on various alternative technical supports. These alternative methods allow for the detection and characterization of CD30 positive cells in different formats, expanding the diagnostic capabilities beyond traditional histology.

For example, in the diagnostic process of Breast Implant-Associated Anaplastic Large Cell Lymphoma (BIA-ALCL), the cytopathological analysis of the effusions associated with a late breast implant plays a pivotal role [19]. It involves the examination of cellular components within the collected fluid sample, which is typically obtained through procedures such as fine-needle aspiration or seroma fluid analysis. These characteristics may include atypical lymphoid cells, large cell size, abnormal cell shape, and distinct cell membrane markers, such as CD30 expression.

Furthermore, in the context of fine-needle aspiration cytology (FNAC), the utilization of specific markers such as CD15, CD30, and PAX5 has proven to be valuable for the identification of Hodgkin lymphoma (HL) and non-Hodgkin large B-cell lymphoma, specifically diffuse large B-cell lymphoma (DLBCL). These markers aid in distinguishing between these subtypes, and their differential expression patterns provide insights into the diagnostic utility of FNAC [20].

### 2.2. Embryonal Carcinoma and CD30

Embryonal carcinoma, a malignant germ cell tumor found primarily in the testes and ovaries, is linked to the expression of CD30. CD30 serves as an invaluable diagnostic marker to detect embryonal carcinoma from other types of germ cell tumors; additionally, its signaling could play a part in embryonal carcinoma cell survival and proliferation similar to its role in other forms of cancer; therefore, more research must be conducted into how CD30 influences embryonal carcinoma cells’ behavior [21].

Immunohistochemistry plays an essential role in accurately classifying and distinguishing malignant germ cell neoplasms from other gynecological and nongynecological tumors, thus having a profound impact on their clinical management. Nuclear markers like SALL4, OCT4, NANOG, and SOX2 are often employed; SOX2 can also be found in the cytoplasm. In addition, membranous markers like CD117, D2-40 CD30 PLAP GP-3 may also be useful; Gp-3 and PLAP proteins have been observed within cells. By combining these markers with non-germ cell tumors markers, such as PLAP and SALL4, it possible to differentiate germ cell tumors from epithelial and gonadal stromal neoplasms in most cases. SALL4 has proven highly specific for germ cell tumors, showing only weak staining on some ovarian clear cell tumors and metastatic gastrointestinal adenocarcinomas, without reacting with any granulosa cell, theca cell, or ovarian stromal cell tumors [22,23].

OCT3/4, NANOG, SOX2, GP-3, and CD30 markers can help reliably differentiate yolk sac tumors and embryonal carcinomas. Positive staining for CD30, OCT3/4, NANOG, SOX2 GP-3, or CD30 tends to occur more commonly among embryonal carcinomas while negative staining occurs more commonly among yolk sac tumors, serving as reliable indicators between them both neoplasms. Furthermore, GP-3 staining is positive in yolk sac tumors but negative in embryonal carcinomas, allowing for further differentiation between them [24,25].

CD30 plays an intricate role in embryonal cancer development and requires further study to understand its specific mechanisms and therapeutic applications within embryonal carcinoma.

### 2.3. Anaplastic Large Cell Lymphoma (ALCL) and CD30

ALCL is an aggressive form of non-Hodgkin lymphoma characterized by large lymphoma cells expressing CD30. CD30 plays a crucial role in its development as it contributes to cell survival, proliferation, and resistance against apoptosis. Furthermore, abnormal activation of CD30 signaling in ALCL often stems from genetic translocations affecting anaplastic lymphoma kinase (ALK) gene translocations that result in hybrid ALK fusion proteins, which further intensify CD30 signaling [26]. Moreover, ALCL ALK+ are more prevalent in children and young adults, with better outcomes, whereas ALCL AKL− occur mostly in adults and elders, with the worst outcome [27].

#### 2.3.1. ALK-Positive (ALK+) Anaplastic Large Cell Lymphoma (ALCL)

An ALK+ anaplastic large cell lymphoma (ALCL) is a subtype of T-cell lymphoma that features large lymphoid cells characterized by their unique, horseshoe-shaped nuclei and ample cytoplasm. ALK+ ALCL can be distinguished from other cancer types by a chromosomal translocation involving ALK, leading to the expression of both ALK protein and CD30 proteins [27]. Noteworthy is another category of ALCL called ALK-negative ALCL, which exhibits similar morphological and phenotypic features but lacks ALK rearrangement and protein expression. Therefore, differentiation diagnosis must also be conducted between ALK+ ALCL and other subtypes of T-cell or B-cell lymphomas that display anaplastic features or CD30 expression.

ALK+ ALCL typically involves both lymph nodes and extranodal sites. Extranodal involvement includes skin, bone, soft tissue, lungs, and liver, and only rarely affecting the gastrointestinal tract or central nervous system. Contrary to classic Hodgkin lymphoma, mediastinal disease is less frequently present with ALK+ ALCL. When investigating bone marrow involvement based on H&E staining, it is estimated to occur in approximately 10% of the cases. Immunohistochemical stains reveal a higher incidence (around 30%) as ALK+ ALCL may involve bone marrow less significantly. Additionally, the small cell variant of ALK+ ALCL may present as leukemic leukemia involving peripheral blood [26,27,28].

ALK+ ALCLs exhibit a diverse array of morphological characteristics. However, all cases contain cells with eccentric, horseshoe-shaped, or kidney-shaped nuclei that often exhibit an eosinophilic region near their nucleus. These cells, commonly called hallmark cells, can be found across all morphologies. While hallmark cells typically feature larger cell sizes with similar cytological features, smaller variants with the same characteristics can also be observed, which aids in accurate diagnosis. Certain cells may appear to contain nuclear inclusions; however, these are invaginations of the nuclear membrane rather than true inclusions and thus they are known as doughnut cells [29].

Morphologically, ALK+ ALCLs range from small cell neoplasms to those dominated by large cells. Multiple patterns have been identified, with the common pattern representing roughly 60% of cases. Tumor cells that display this pattern feature abundant cytoplasms that may appear clear, basophilic, or eosinophilic. Multiple nuclei arranged in an arc-shaped configuration give rise to cells resembling Reed–Sternberg cells. Nuclear chromatin tends to be finely clumped or dispersed, with multiple small, basophilic nucleoli. When composed of larger cells, nucleoli may become more prominent; however, eosinophilic or inclusion-like nucleoli are rare. When lymph node architecture remains partially preserved, tumor growth typically occurs within its sinuses, resembling metastatic tumor growth [30].

The lymphohistiocytic pattern, found in approximately 10% of cases, is characterized by tumor cells mixed with an abundance of reactive histiocytes that may obscure any malignant cells present, leading to misdiagnosis as reactive lesions. Neoplastic cells found here tend to be smaller than in the common pattern but tend to cluster around blood vessels; CD30 and ALK antibodies may help identify and highlight them more readily than others can; histiocytes may even exhibit evidence of red blood cell engulfment (erythrophagocytosis) [31].

The small-cell pattern, accounting for 5–10% of cases, typically feature small to medium-sized neoplastic cells with irregular nuclei, and pale cytoplasm and centrally located nuclei that have become cancerous over time. This is the origin of the cells’ “fried egg cells” nickname. Rarely, signet ring-like cells may also be observed. Hallmark cells, characteristic of ALCL, are always present and often cluster around blood vessels. This variant can often be misdiagnosed during the traditional examination as peripheral T-cell lymphoma not otherwise specified (NOS). Smear preparations from peripheral blood samples may reveal small, atypical cells with folded nuclei that resemble flower petals and occasional large cells with vacuolated, blue cytoplasm. The Hodgkin-like pattern, found in approximately 3% of cases, displays nodular sclerosis typical for Hodgkin lymphoma. A total of 15% of lymph node biopsies may show multiple patterns simultaneously—known as composite patterns—on one biopsy sample; it is important to remember that relapses may reveal different characteristics than the initial presentation. For example, ALCL tumors display cells characterized by monomorphic, round nuclei either predominantly or mixed with more pleomorphic nuclei, either alone or mixed together with more pleomorphic ones. Some cases may include high proportions of multinucleated giant cells or display sarcomatous features. Occasionally, the tumor may exhibit a hypocellular appearance characterized by myxoid or edematous background. Myxoid tumors can sometimes present themselves, with spindle cells mimicking sarcoma while occasional rare cases containing clusters of malignant cells within an otherwise reactive lymph node are present; capsular fibrosis combined with tumor nodules can also be seen, suggesting metastatic non-lymphoid malignancies [31,32,33,34]. 

#### 2.3.2. ALK-Negative Anaplastic Large Cell Lymphoma (ALCL)

An anaplastic large cell lymphoma (ALCL) that does not express detectable amounts of the ALK protein is known as an ALK− anaplastic large cell lymphoma (ALCL). Morphologically, ALK− ALCL cannot reliably be differentiated from its counterpart ALK+ ALCL on appearance alone; it was initially included as a provisional entity in the 2008 edition of WHO classification but is now recognized as an established one. Therefore, ALK− ALCL should be distinguished from conditions like primary cutaneous ALCL (C-ALCL), various subtypes of CD30+ T-cell or B-cell lymphoma subtypes exhibiting anaplastic features, or classic Hodgkin lymphoma (CHL).

Anaplastic large cell lymphoma (ALCL) of the ALK-negative (ALK−) type typically affects both lymph nodes and extranodal tissues; however, extranodal involvement tends to be less frequent than ALK+ ALCL cases. Locations most likely to be impacted include bone, soft tissue, and skin. As it is crucial to differentiate between ALK− ALCL cases from primary cutaneous ALCL (C-ALCL) and those involving the digestive tract from CD30+ enteropathy-associated lymphomas or other intestinal T-cell lymphomas, which can be difficult to do based on lymph nodes alone, it may be useful to explore patient histories for any associated lesions; this could indicate nodal involvement by C-ALCL primary cells.

An ALK-negative anaplastic large cell lymphoma (ALCL) typically affects lymph nodes or tissues, with solid sheets of cancerous cells replacing their original architecture. When the lymph node architecture remains undisturbed, cancerous cells tend to collect in sinuses or T-cell areas, forming a cohesive pattern resembling carcinoma. The absence of these features should raise suspicion for peripheral T-cell lymphoma (PTCL, NOS). Unfortunately, needle biopsies may not provide sufficient material to assess these architectural features. Overall, ALK− ALCL typically exhibits features similar to those described for ALK+ ALCL; no unique morphological patterns have been noted. However, features like sclerosis and eosinophils should raise suspicion for classic Hodgkin lymphoma (CHL). Cases resembling CHL with confirmed T-cell origin should be classified as PTCL NOS. Cases mimicking CHL can also indicate nodal involvement from lymphomatoid papulosis. Cytologically, ALK− ALCL cells demonstrate similarities with ALK+ ALCL in terms of appearance; however, small tumor cells typically found in ALK+ ALCL variants involving small-cell and lymphohistiocytic subtypes should not be evident in ALK− ALCL cases. Biopsies of ALK− ALCL typically reveal large, pleomorphic cells with prominent nucleoli. Multinucleated and wreathlike cells may also be present; mitotic figures are not uncommon. On occasion, hallmark cells with eccentric, horseshoe-shaped, or kidney-shaped nuclei may also be observed to varying degrees. Compared with classic ALK+ ALCL cells, ALK− ALCL neoplastic cells often show greater size, more pleomorphism, and an elevated nuclear-to-cytoplasmic ratio (N: C ratio). An increased N: C ratio may suggest the possibility of PTCL and NOS; however, in such a condition, there will likely be an admixture of abnormal small to medium-sized lymphocytes with an overall uniformly neoplastic cell population, and the sheet-like or sinus pattern of infiltration characteristic of ALCL is absent [35,36,37,38].

Additionally, CD30 positive cells are also present in another rare pathology, defined as an anaplastic lymphoma kinase (ALK) negative. Breast implant-associated anaplastic large-cell lymphoma (BIA-ALCL) typically involves the accumulation of fluid around the implant site subsequent to the placement of textured surface breast implants during either aesthetic or reconstructive procedures, which may be accompanied by breast swelling, asymmetry, or pain. Dermatological indications (such as redness, small, raised bumps) and localized swelling of the lymph nodes on one side have been documented [39]. Constitutional symptoms like elevated body temperature, swelling of the lymph nodes, night sweats, and exhaustion can also coexist [40].

### 2.4. Mycosis Fungoides and CD30

Mycosis fungoides (MF), a rare type of non-Hodgkin’s lymphoma, are usually considered an indolent CD30 negative cutaneous T-cell lymphoma, but some patients pursue a strong and assertive direction [41]. The expression of CD30 has been observed in approximately 40% of cases with histologic transformation in mycosis fungoides (MF) [42]. Accordingly, before C-ALCL, which is distinguished by a substantial presence of CD30+ cells, can be identified, the potential for transformed MF needs to be excluded.

In general, the disease primarily affects the skin, displaying diverse distribution patterns for an extended period of time. In advanced stages, there is a possibility of the disease spreading to extracutaneous sites, predominantly involving the lymph nodes, liver, spleen, lungs, and bloodstream [43]. The involvement of the bone marrow is uncommon [44].

In the early phases, the skin manifestations are commonly localized to sun-protected regions. Patients diagnosed with tumor-stage mycosis fungoides typically present with a combination of patches, plaques, and tumors, often accompanied by ulceration [44].

The histological appearance of the skin lesions varies depending on the stage of the disease. In the early stages, patch lesions exhibit superficial infiltrates that resemble bands or lichen, mainly composed of lymphocytes and histiocytes. Atypical cells with small to medium-sized, highly indented (cerebriform) nuclei are infrequent and mostly confined to the epidermis. These cells tend to occupy the basal layer of the epidermis and can be seen as individual cells or arranged in a linear pattern, often surrounded by a halo. In typical plaque lesions, there is a more pronounced tendency for the atypical cells to invade the epidermis (epidermotropism). Pautrier microabscesses, which are collections of atypical cells within the epidermis, are a distinctive feature but are observed in only a minority of cases [45]. As the disease progresses to the tumor stage, the infiltrates in the dermis become more diffuse, and the epidermotropism may diminish. The tumor cells increase in both number and size, exhibiting varying proportions of small, medium-sized, and large cerebriform cells with nuclei that display pleomorphic or blastoid characteristics. Histological transformation, defined by the presence of more than 25% large lymphoid cells in the dermal infiltrates, may occur, particularly in the tumor stage [46]. These large cells can be either CD30-negative or CD30-positive.

Enlarged lymph nodes from patients with mycosis fungoides frequently show dermatopathic lymphadenopathy with paracortical expansion due to the large number of histiocytes and interdigitating cells with abundant, pale cytoplasm. The International Society for Cutaneous Lymphomas (ISCL) I European Organisation for Research and Treatment of Cancer (EORTC) staging system for clinically abnormal lymph nodes (>1.5 cm) in mycosis fungoides and Sezary syndrome recognizes three categories: N1, reflecting no involvement; N2, early involvement (with no architectural effacement); and N3, overt involvement (with partial or complete architectural effacement) [47].

## 3. Therapeutic Approaches Based on CD30

Based on the information outlined thus far, it is evident that CD30 plays an essential role in tumor development and progression. As such, we need to determine whether it can be targeted with cytostatic treatments; herein, we explore several promising approaches currently under study.

### 3.1. Brentuximab Vedotin

According to current guidelines from both ESMO (European Society for Medical Oncology, Lugano, Switzerland) and NCCN (National Comprehensive Cancer Network, Plymouth Meeting, PA, USA), patients diagnosed with classical Hodgkin lymphoma (HL) should undergo chemotherapy or combined modality therapy followed by involved-field radiotherapy for initial treatment. Once finished, the disease should be restaged to evaluate its status; treatment intensity will depend on the clinical stage and any risk factors present; risk-adapted therapy enables excellent cure rates regardless of the stage at diagnosis.

Recently, ADCs (antibody-drug conjugates) have revolutionized cancer treatments such as classical Hodgkin lymphoma (HL), systemic anaplastic large cell lymphoma (sALCL), and breast cancer. ADCs represent an innovative advance by targeting tumor cells while limiting damage to normal tissues. Such ADC is brentuximab vedotin (ADCETRIS), specifically targeting the CD30 membrane receptor. CD30 is an ideal target for ADC therapy due to its high expression levels in cancerous cells like those seen in classical HL and sALCL while being limitedly expressed on normal cells (primarily activated B cells and T cells). Furthermore, CD30 expression remains consistent among HL and sALCL cells regardless of disease stage, therapy history, or transplant status. Therefore, anti-CD30 monoclonal antibodies could be used in combination with other cytotoxic agents for patients with CD30-positive Hodgkin lymphoma [48].

At times, even after receiving initial therapy, many patients relapse or develop resistance. When this occurs, brentuximab vedotin should be considered as a potential option for patients with recurrent classical Hodgkin lymphoma (HL) who have not responded to high-dose chemotherapy and autologous stem cell transplantation (HDCT/ASCT) or two prior chemotherapy regimens regardless of eligibility for HDCT/ASCT [48,49,50,51].

Intravenous brentuximab vedotin has proven effective as consolidation therapy after autologous stem cell transplantation (ASCT) and salvage therapy in CD30-positive classical Hodgkin lymphoma (HL) cases. Brentuximab vedotin significantly extends progression-free survival (PFS) compared to the placebo in the phase 3 AETHERA trial as post-ASCT consolidation therapy. Benefits observed during the primary analysis with a median follow-up timeframe of 30 months included durable responses, lasting about one-year post-primary analysis. In a pivotal phase 2 trial, approximately 75% of patients receiving brentuximab vedotin as salvage therapy achieved an objective response and met its primary endpoint [52].

Estimated median progression-free survival (PFS) was estimated at 5.6 months; the median overall survival after 18.5 months follow-up was 22.4 months. Importantly, responses to brentuximab vedotin salvage therapy were long-lasting: estimates for 5-year overall survival and PFS rates stood at 41% and 22%, respectively, in the general patient population. In addition, retrospective analyses of real-world data provide further proof of brentuximab vedotin’s efficacy as salvage therapy, and retreatment with brentuximab vedotin was proven successful for patients who achieved complete or partial responses during initial treatment but then experienced relapses [53].

Salvage therapy with brentuximab vedotin has demonstrated promising results in phase 2 trials in real-world settings in treating patients with CD30-positive classical Hodgkin lymphoma (HL). In addition, these therapies had shown high objective response rates for retreatment of patients who initially responded well to brentuximab vedotin therapy but later relapsed; moreover, positive responses have proven sustainable during long-term follow-up. As consolidation therapy following autologous stem cell transplantation (ASCT), brentuximab vedotin has proven its efficacy at prolonging progression-free survival (PFS) compared to placebo, as per primary analysis results. In addition, Brentuximab vedotin therapy has provided long-term positive results maintained over time, while maintaining acceptable tolerability and safety profiles; most adverse events could be managed through dose adjustments and/or postponements. Given the limited treatment options for those living with relapsed or refractory HL, brentuximab vedotin represents an effective option for patients who have failed high-dose chemotherapy and ASCT or at least two prior chemotherapy regimens. Furthermore, it can serve as post-ASCT consolidation therapy in cases at increased or high risk for disease progression or recurrence post-ASCT [54,55] (Table 2).

### 3.2. Bispecific Antibodies (bsAbs)

NK cells are an integral component of the innate immune system and play an essential role in immunosurveillance by targeting tumor cells. Unlike other immune cells, however, NK cells do not possess antigen-specific receptors; instead, they utilize various activation and effector functions mediated by multiple receptors, including CD16a (also known as FcgRIII), which facilitates antibody-dependent cytotoxicity (ADCC).

To utilize the cytotoxic capabilities of natural killer (NK) cells, a bispecific antibody (bsAb) was developed to target CD30/CD16. The HRS-3/A9 bsAb contains two arms; one attaches directly to the CD30 antigen binding site, while the other recruits NK cells via binding to its CD16 receptor site. This bispecific construction facilitates the formation of an “immunological synapse”, leading to the activation and release of cytotoxic substances, such as granzyme B, and perforin release and subsequent cell death of CD30+ cells. Preclinical studies using SCID mice carrying CD30+ human Hodgkin lymphoma tumors confirmed this effect by complete remission [56,57,58]. The mechanism of action of bispecific antibodies is presented in Figure 2.

Clinical trials conducted with Hodgkin lymphoma patients who had become resistant to treatment with HRS-3/A9 resulted in 9 out of 15 developing antidrug antibodies (ADA). These adverse drug reactions may compromise treatment efficacy and safety profiles. Four patients experienced allergic reactions upon reinfusion and were therefore excluded from further care. Due to these limitations and challenges, further development of this drug was abandoned, while efforts were instead directed toward searching for more innovative bispecific antibodies capable of activating NK cells [57].

## 4. Conclusions

CD30 plays an extraordinarily diverse role in oncogenesis, which is yet to be fully understood. Its importance is currently well-established as an immunohistochemical marker used for diagnosing various lymphoid malignancies, such as classic Hodgkin lymphoma or ALCL, as well as non-lymphoid malignancies, such as embryonal carcinoma. Furthermore, CD30 can be considered a valuable therapeutic target for aggressive lymphoid malignancies, resistance to other therapeutic lines, and for consolidation therapy to prolong progression-free survival. In addition to its high efficacy, CD30 may be particularly useful for cancer treatment since it is strongly expressed by malignant cells but has a low expression in normal cells, which reduces its adverse side effects.

## Figures and Tables

**Figure 1 cells-12-01783-f001:**
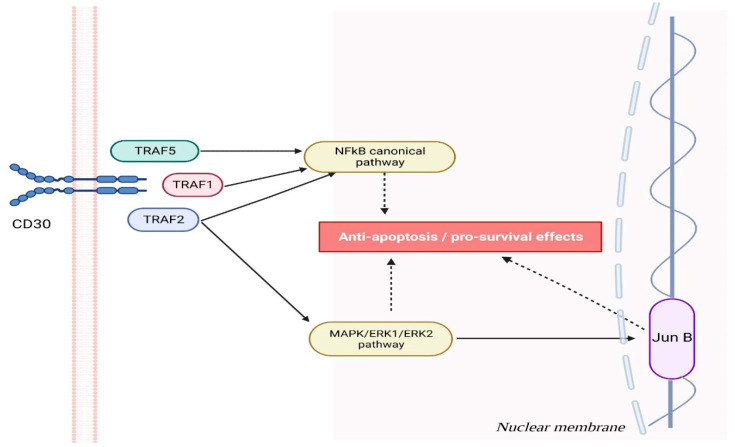
The CD30 molecule plays a critical role in cell survival by engaging various signaling pathways that confer an advantage to cells with higher CD30 levels. When activated, CD30 forms trimers and sends out signals through tumor necrosis factor receptor-associated proteins (TRAF), specifically TRAF2, TRAF1, and TRAF5, to stimulate nuclear factor-kappa B (NFkB) pathways and thus increase survival chances. CD30 binding initiates signaling through mitogen-activated protein kinase (MAPK) pathways, including ERK1 and ERK2, that activate mitogen-activated protein kinase kinase kinase. This provides cancerous cells with various anti-apoptotic and pro-survival benefits, with JunB acting as a nuclear transcription factor to further support survival while upregulating CD30 expression.

**Figure 2 cells-12-01783-f002:**
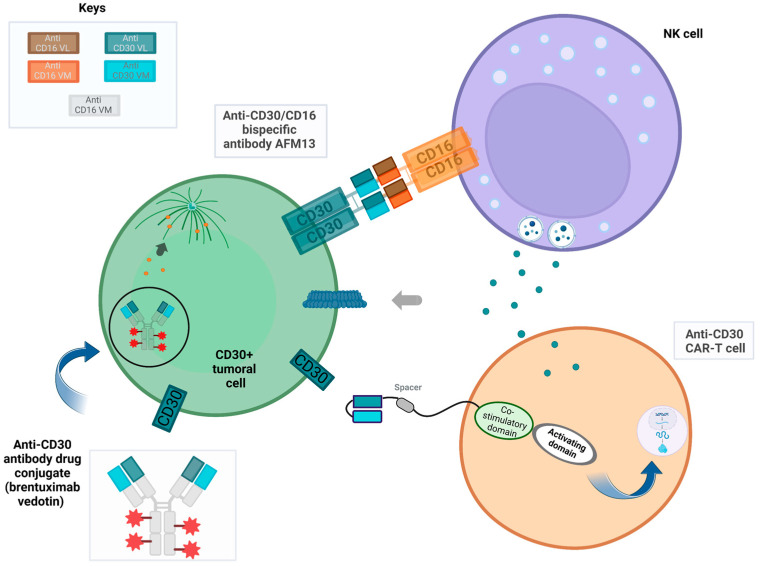
AFM13 (bispecific antibody), targeting both CD30 antigen and CD16A on NK cells, plays an essential role in creating an “immunological synapse”. This interaction leads to activation of NK cells which then release cytotoxic substances like granzyme B and perforin, which result in cell lysis of CD30 positive cells. Brentuximab vedotin targets CD30-positive cells specifically, leading to their receptor internalization. When exposed to intracellular lysozymes, it releases MMAE, which interferes with tubulin formation and triggers cell apoptosis. Furthermore, when an anti-CD30 CAR-T cell binds its target, downstream signaling pathways and costimulatory signals become active, inducing T-cell effector functions including perforin and granzyme release by T cells that leads to their ultimate destruction and demise of CD30-positive cells.

**Table 1 cells-12-01783-t001:** Differential diagnostics of HD with subtypes, ALCL systemic vs. primary cutaneous, and MF CD30+.

		Clinical Symptoms (Skin, Lymph Nodes, Internal Organs)	Histopathology	Immunohistochemistry
Lymphomas that affect the thymus (T-cell, B-cell lineages and Hodgkin lymphomas)	Nodular Sclerosis (NS) (subtype of Hodgkin lymphomas—cHLs)	Enlarged lymph nodes, particularly in the neck, chest, or armpits; Pain or discomfort in the affected lymph nodes after consuming alcohol. B symptoms: Fever, night sweats, and unexplained weight loss. Fatigue and general malaise. Itchy skin (pruritus).	- large sclerotic masses with foci of necrosis and eosinophilic abscesses; Reed–Sternberg cells (RS) - large with abundant eosinophilic cytoplasm, - large double or multiple nuclei, eosinophilic nucleoli; Lacunar cells (LC) - cells more frequently associated to cHL-NS; - small hyperlobulated nuclei, small nucleoli; - clear, retracted cytoplasm.	CD30 + (85–96%, Memb + Golgi area) CD15 + (75–85%) CD20 + (<20–40%) CD79a (Rarely+) PAX5; OCT2; MUM1; EBER (EBV); LMP-1 (EBV) Cyclin E Fascin
	PMBL (primary mediastinal B-cell lymphoma)	Limited to the thorax with no involvement of lymph nodes or other lymphoid organs (only supraclavicular nodes are eventually reached); Chest pain or pressure due to the tumor’s location in the mediastinum; Difficulty breathing or shortness of breath; Coughing or wheezing.	bulky, solid masses of >10 cm, with local symptoms of rapid growth, invasion, and compression of vital structures. Neoplastic cells - variable size, sometimes with pale clear cytoplasm in the central part of the tumor; - peripherally distributed lymphocytes in the sclerotic background.	CD30 + (weakly or focally expressed, with lesser intensity than in cHL) CD45 CD20 CD79a PAX5; OCT2; BOB1; MUM1 MAL; CD23; BCL6; P63
	T-lymphoblastic lymphoma (T-Lb)	Enlarged lymph nodes, commonly in the neck, chest, or armpits. Swelling or mass in the thymus or mediastinum. Difficulty breathing or shortness of breath. Coughing or wheezing.	- thymic remnant and epithelial cell (EC) networks are rarely found; - frequent mitosis and extensive necrosis.	CD30 (rare) T-cell markers: CD3, CD5, and CD7 TdT CD1a; CD99; CD4 and CD8; cyclin-dependent kinase-6 (CDK6)
	B1—Thymoma	Asymptomatic: Many B1 thymomas are discovered incidentally during imaging studies performed for other reasons. Chest pain or discomfort. Coughing or respiratory symptoms (if the tumor compresses nearby structures). Difficulty breathing or shortness of breath.	- EC network may partially be seen even in necrotic tumors; - frequent mitosis and extensive necrosis	(+) in >90%: CK5/6, CK7, CK8, CK18, CK19, p63 (+) in 50–90%: CD15, CD57 (leu7), PAX-8 Cytokeratins, CD30
ALCL	Anaplastic large cell lymphoma, ALK positive	- involves both lymph nodes and extranodal sites (skin, bone, soft tissue, lungs, liver + rarely, gastrointestinal tract or central nervous system);	- chromosomal translocation involving ALK, leading to the expression of both ALK protein and CD30 proteins; Tumor cells - abundant cytoplasm that may appear clear, basophilic, or eosinophilic; - multiple nuclei arranged in an arc-shaped configuration; - nuclear chromatin tends to be finely clumped or dispersed, with multiple small, basophilic nucleoli. Hallmark cells - eccentric, horseshoe-shaped, or kidney-shaped nuclei that often exhibit an eosinophilic region near their nucleus; - ample cytoplasm; Doughnut cells - invaginations of the nuclear membrane; “Fried egg cells” - small to medium-sized neoplastic cells with irregular nuclei, and pale cytoplasm and centrally located nuclei that have become cancerous over time	(+) in >90% ALK, CD30, clusterin, CD43, cytotoxic molecules (TIA-1, perforin, granzyme B) (+) in 50–90% CD2, CD4, CD25, CD45, EMA, galectin-3 (+) in 10–50% CD3, CD5, CD7, CD15, fascin, bcl-6 (+) in <10% CD8, CD20, CD28, PAX-5
	Anaplastic large cell lymphoma, ALK negative	- affects both lymph nodes and extranodal tissues; - extranodal involvement tends to be less frequent compared to ALK+ ALCL cases (bone, soft tissue, and skin);	- large lymphoid cells with horseshoe-shaped nuclei and ample cytoplasm; - does not express detectable amounts of the ALK protein; - similar features to ALCL ALK positive.	(+) in >90% CD30, clusterin, CD43, cytotoxic molecules (TIA-1, perforin, granzyme B) (+) in 50–90% CD2, CD4, CD25, CD45, EMA, galectin-3 (+) in 10–50% CD3, CD5, CD7, CD15, fascin, bcl-6 (+) in <10% ALK, CD8, CD20, CD28, PAX-5
Mycosis fungoides		- in general => skin manifestations: initially are localized to sun-protected regions, then a combination of patches, plaques, and tumors +/− ulceration - in advanced stages => extracutaneous sites (lymph nodes, liver, spleen, lungs, and bloodstream); +(rarely) the bone marrow.	- early stages: superficial infiltrates that resemble bands or lichen, mainly composed of lymphocytes and histiocytes; - atypical cells, with highly indented (cerebriform) nuclei that are infrequent and mostly confined to the epidermis; - Pautrier microabscesses (in a minority of cases).	(+) in >90% CD2, CD3, CD5, CD4, CD45RO (+) in 10–50% CD7. CD30 (+) in <10% CD8, CD25

**Table 2 cells-12-01783-t002:** Brentuximab vedotin’s function in various lymphomas.

	Brentuximab Vedotin’s Function in Various Lymphomas
Classical Hodgkin lymphoma	- Option for patients with recurrent classical Hodgkin lymphoma that did not respond to high-dose chemotherapy and autologous stem cell transplantation (HDCT/ASCT) or two prior chemotherapy regimens regardless of eligibility for HDCT/ASCT; - intravenous brentuximab vedotin = in consolidation therapy after ASCT and salvage therapy for CD30-positive HL.
ALCL	- approved for the treatment of systemic ALCL in adult patients who have not responded to prior multi-agent chemotherapy or who are not candidates for ASCT; - is not typically used as a first-line treatment for primary cutaneous ALCL.
Mycosis Fungoides	- brentuximab vedotin may be utilized in the treatment of MF with CD30 expression when it is refractory to other treatments or in cases of relapsed disease.

## Data Availability

No new data were created or analyzed in this study. Data sharing is not applicable to this article.

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
