# Peer review of "The Polyvalent Role of CD30 for Cancer Diagnosis and Treatment"

_cells, 2023, doi:10.3390/cells12131783_

Round 1
Reviewer 1 Report
Dear Authors,
I read the article with great interest and I found it valuable and worth of publication but after minor revisions. Brentuximab vedotin is widely used in every day treatment of CD30+ lymphomas and Hodgkin disease. More detailed information concerning primary cutaneous ALCL as well as CD30+ transformation of other cutaneous T cell lymphoma: Mycosis fungicides (MF) have been missing.
I would like to ask for adding the table with differential diagnostics of HD with subtypes, ALCL systemic vs primary cutaneous and MF CD30+ ( skin , lymph nodes, internal organs - clinical symptoms, histopathology, immunohistochemistry). It would be interested to add the 2nd table: with place of brentuximab vedotin in treatment of HD - with subtypes, ALCL systemic vs primary cutaneous and MF CD30+ ( ex. with information concerning the line of treatment according to the reccommendations). These changes will make reading easier and the information more practical.
Best reagards
Author Response
Thank you for your report! In response to your thoughtful feedback, we have carefully considered each of your points and have addressed them in a point-by-point manner below:
Comment: We haven’t given more information about CD30+ transformation of other cutaneous T cell lymphoma, such as mycosis fungicides.
Response: Thank you for your suggestion! In consideration of that, we elaborated on mycosis fungicides and its major characteristics (histopathology, clinical symptoms etc.) on subchapter 2.4, right after ‘2.3. Anaplastic large cell lymphoma (ALCL) and CD30’.
Comment: Our manuscript lacks of tables that could make reading easier.
Response: Thank you for your suggestion! Tables would certainly enlighten the content of the article and make it easier to read. We have added two tables. Table 1 describes the major aspects of the lymphoma elaborated throughout the article, including clinical symptoms, histopathology, and immunohistochemistry, while Table 2 expounds how Brentuximab vedotin acts in therapies for different pathologies.
Reviewer 2 Report
The manuscript “The Polyvalent Role of CD30 for Cancer Diagnosis and Treatment” is an interesting review on the molecular structure, biological meaning, and phenotypical expression of CD30 in different neoplasms exploring the ongoing research into the potential use of CD30-targeted therapies for autoimmune disorders. The review is sufficiently timely and exhaustive, the manuscript is well prepared.
Major points: Almost all the CD30 positive neoplasms are composite tumors in which the CD30 component is a part of corresponding tumors even minimal in some entities (i.e. Hodgkin lymphoma). Any possible diagnostic and predictive speculation cannot work regardless of the morphological identification of CD30 positive cells and their quantification. This point should be evaluated and discussed.
CD30 positive cells are present in breast implant-associated anaplastic large cell lymphoma and effusions. Whereas this is a rare entity it should be mentioned and briefly described at least.
CD30 positive cells can be identified on different technical supports other than on conventional histological sections. Different reports have enhanced this aspect; below are just some of them. As the present study is a comprehensive review, this aspect and some of corresponding reports should be reported.
Julien LA, et al. Breast implant-associated anaplastic large cell lymphoma and effusions: A review with emphasis on the role of cytopathology. Cancer Cytopathol. 2020 Jul;128(7):440-451.
Cozzolino I, et al.: CD15, CD30, and PAX5 evaluation in Hodgkin's lymphoma on fine-needle aspiration cytology samples. Diagn Cytopathol. 2020 Mar; 48(3):211-216.
The authors assess that it can be difficult to differentiate T-Lymphoblastic Lymphoma and B1 thymoma using lymphoblasts alone and that lymphoblastic precursors like LMO2 can often help on the differential diagnosis. This sentence may be acceptable whereas single antigens and corresponding antibodies rarely can discriminate between these two entities. Instead, a complete and wide phenotypic profile and or molecular testing can more probably succeed in the differential diagnosis.
The manuscript “The Polyvalent Role of CD30 for Cancer Diagnosis and Treatment” is an interesting review on the molecular structure, biological meaning, and phenotypical expression of CD30 in different neoplasms exploring the ongoing research into the potential use of CD30-targeted therapies for autoimmune disorders. The review is sufficiently timely and exhaustive, the manuscript is well prepared.
Major points: Almost all the CD30 positive neoplasms are composite tumors in which the CD30 component is a part of corresponding tumors even minimal in some entities (i.e. Hodgkin lymphoma). Any possible diagnostic and predictive speculation cannot work regardless of the morphological identification of CD30 positive cells and their quantification. This point should be evaluated and discussed.
CD30 positive cells are present in breast implant-associated anaplastic large cell lymphoma and effusions. Whereas this is a rare entity it should be mentioned and briefly described at least.
CD30 positive cells can be identified on different technical supports other than on conventional histological sections. Different reports have enhanced this aspect; below are just some of them. As the present study is a comprehensive review, this aspect and some of corresponding reports should be reported.
Julien LA, et al. Breast implant-associated anaplastic large cell lymphoma and effusions: A review with emphasis on the role of cytopathology. Cancer Cytopathol. 2020 Jul;128(7):440-451.
Cozzolino I, et al.: CD15, CD30, and PAX5 evaluation in Hodgkin's lymphoma on fine-needle aspiration cytology samples. Diagn Cytopathol. 2020 Mar; 48(3):211-216.
The authors assess that it can be difficult to differentiate T-Lymphoblastic Lymphoma and B1 thymoma using lymphoblasts alone and that lymphoblastic precursors like LMO2 can often help on the differential diagnosis. This sentence may be acceptable whereas single antigens and corresponding antibodies rarely can discriminate between these two entities. Instead, a complete and wide phenotypic profile and or molecular testing can more probably succeed in the differential diagnosis.
Author Response
Thank you for your report! In response to your thoughtful feedback, we have carefully considered each of your points and have addressed them in a point-by-point manner below:
Comment 1: We should discuss in our article how CD30-positive neoplasms are composite tumors, so any possible diagnostic and predictive speculation cannot work since the lymphoma consist of different cell types or components.
Response: Thank you for your feedback! We agree that it is important to consider the composite nature of these tumors. As you rightly pointed out, in many cases, the CD30 component is just a small part of the overall tumor, particularly in entities like Hodgkin lymphoma. In response to your comment, we evaluated and discussed the limitations of relying solely on CD30 positivity for diagnostic and predictive purposes. We addressed the composite nature of CD30 positive neoplasms and the potential implications this has on accurate interpretation and clinical decision-making.
Comment 2: Since CD30 positive cells are also present in breast implant-associated anaplastic large cell lymphoma and its effusions, we should have mentioned and described this pathology.
Response: Thank you for your insightful comment! It is indeed important to mention BIA-ALCL as a CD30 positive pathology. Since it’s an anaplastic lymphoma kinase (ALK) negative one we mentioned and described it at the subchapter entitled ‘2.3.2. ALK-negative anaplastic large cell lymphoma (ALCL)’.
Comment 3: We should have reported that, besides the traditional histology, there are other technical supports that are being used in order to allow for the detection and characterization of CD30 positive cells.
Response: Thank you for your suggestion! We have read the reports highlighting information about the identification of CD30 positive cells on different technical supports beyond conventional histological sections. By including information about this aspect, we can provide a more well-rounded understanding of the topic and accommodate the interests of a wider readership.
In response to your comment, we ensured that the manuscript incorporates a section dedicated to discussing the different technical supports available for identifying CD30 positive cells. We included relevant reports that have enhanced our understanding of this aspect, as suggested.
Comment 4: We should provide a complete description of both T-Lymphoblastic Lymphoma and B1 thymoma, in order to enhance the differences between them.
Response: Thank you for your recommendation! In consideration of that, we provided a table (Table 1) that describes the major elements of each major lymphoma mentioned throuughout the article.